# Structural conservation in a membrane-enveloped filamentous virus infecting a hyperthermophilic acidophile

Ying Liu[1], Tomasz Osinski [2], Fengbin Wang [2], Mart Krupovic [1], Stefan Schouten[3,4], Peter Kasson[5], David Prangishvili[1] & Edward H. Egelman [2]

Different forms of viruses that infect archaea inhabiting extreme environments continue to be discovered at a surprising rate, suggesting that the current sampling of these viruses is sparse. We describe here *Sulfolobus* filamentous virus 1 (SFV1), a membrane-enveloped virus infecting *Sulfolobus shibatae*. The virus encodes two major coat proteins which display no apparent sequence similarity with each other or with any other proteins in databases. We have used cryo-electron microscopy at 3.7 Å resolution to show that these two proteins form a nearly symmetrical heterodimer, which wraps around A-form DNA, similar to what has been shown for SIRV2 and AFV1, two other archaeal filamentous viruses. The thin (∼ 20 Å) membrane of SFV1 is mainly archaeol, a lipid species that accounts for only 1% of the host lipids. Our results show how relatively conserved structural features can be maintained across evolution by both proteins and lipids that have diverged considerably.

[1] Institut Pasteur, Department of Microbiology, 25 rue du Dr. Roux, Paris 75015, France. [2] Department of Biochemistry and Molecular Genetics, University of Virginia, Charlottesville, VA 22908, USA. [3] NIOZ Royal Netherlands Institute for Sea Research, Texel 1790 AB, The Netherlands. [4] Department of Marine Microbiology and Biogeochemistry, Utrecht University, Utrecht 3508 TC, The Netherlands. [5] Departments of Molecular Physiology and Biological Physics and Biomedical Engineering, University of Virginia, Charlottesville, VA 22908, USA. These authors contributed equally: Ying Liu, Tomasz Osinski, Fengbin Wang. Correspondence and requests for materials should be addressed to D.P. (email: david.prangishvili@pasteur.fr) or to E.H.E. (email: egelman@virginia.edu)

Viruses hold immense interest in areas ranging from the origin of life to biotechnological questions of how to package nucleic acids for delivery to cells. In addition, many viruses such as HIV, influenza and Ebola pose urgent threats to human health. Viruses that infect humans are intensively studied for obvious reasons, while bacteriophages (viruses that infect bacteria) have been an important tool in molecular biology, and more recently have re-emerged as promising therapeutic agents against antibiotic-resistant bacteria[1]. However, we know far less about viruses that infect archaea, the third domain of life. New forms of such viruses continue to emerge at a surprising rate[2,3], suggesting that our sampling of such viruses remains quite sparse. In this paper we describe a novel filamentous double-stranded (ds) DNA virus that infects a host living in nearly boiling acid, and show how components of the virion (proteins and lipids) can diverge considerably while preserving the overall virion organization and mode of the dsDNA packaging.

The simplest structural solution for packaging of a linear double-stranded DNA (dsDNA) molecule into a virion appears to be its condensation into a filamentous nucleoprotein helix. This would be similar to the packaging of a single-stranded RNA genome in tobacco mosaic virus, the first virus ever isolated[4,5]. Surprisingly, such virion organization is rare among dsDNA viruses. Indeed, filamentous viruses with linear dsDNA genomes are known to parasitize only hyperthermophilic members of the phylum Crenarchaeota in the domain Archaea[6]. They have been unified into the families *Rudiviridae*, *Lipothrixviridae*, and *Tristromaviridae*, the virions of which carry one, two, or three major coat proteins (MCPs), correspondingly[7]. Members of the *Rudiviridae* are non-enveloped whereas members of the two other families are covered with lipid-containing envelopes. The three-dimensional reconstructions of the rudivirus SIRV2 (*Sulfolobus islandicus virus 2*) and the lipothrixvirus AFV1 (*Acidianus filamentous virus 1*), both at near-atomic resolution, revealed unexpected similarities in the virion organization of the members of the two families[8,9]. In both cases the nucleoprotein helix is composed of asymmetric units which contains two molecules of the MCPs; a homodimer in the case of SIRV2 and a heterodimer in the case of AFV1. Moreover, in both virions DNA was found in A-form, although with different packaging parameters. With the aim of better understanding the evolutionary history of filamentous dsDNA viruses and general mechanisms of virus

evolution, we have isolated and characterized a new species of hyperthermophilic crenarchaeal viruses — *Sulfolobus* filamentous virus 1, and report here on its structural and genomic properties.

## Results

**Virus and host isolation**. The replication of the filamentous virus-like particles (VLPs), with lengths of about 900 nm, was observed in the enrichment culture established from an environmental sample collected at hot, acidic spring Umi Jigoku in Beppu, Japan (75 °C, pH 3.6). From the enrichment culture, 150 single cell isolates were colony purified. Their ability to replicate the VLPs was verified by transmission electron microscopy (TEM) observations of "infected" cell cultures. Three VLP-replicating strains were identified. One of them, designated as BEU9, was subjected to an additional round of colony purification. The 16S rRNA gene sequence of the isolate revealed that it represents a strain of *Sulfolobus shibatae*, with 99% sequence identity to *Sulfolobus shibatae* B12[10]. The virus replicating in *S. shibatae* BEU9 was isolated from a single plaque and named *Sulfolobus* filamentous virus 1 (SFV1) (Fig. 1).

In order to determine the host range of SFV1, virus replication was analyzed in collection strains of the genus *Sulfolobus*, including *S. islandicus* strains REY15A[11], HVE10/4[11], and LAL14/1[12], *S. solfataricus* strains P1 (GenBank accession no. NZ_LT549890), P2[12] and 98/2[13], and *S. acidocaldarius* strain DSM639[14]. None of these strains served as a viral host.

**Virion morphology**. From images of negatively stained samples (Fig. 1a, b), virions of SFV1 appear as flexible filamentous particles that are 845 ± 15 nm long. Each end of the virion is tapered, measures 110 ± 12 nm in length (Supplementary Fig. 1a), and consists of a "neck" and a mop-like structure with several thin, short fibers (Fig. 1b). Intriguingly, in the virus preparations we frequently observed virions with significantly elongated "necks" at one or both termini, which could reach 700 nm in length (Fig. 1b and Supplementary Fig. 1b).

**Viral genome**. The nucleic acid extracted from the purified SFV1 virions was insensitive to RNase A but could be digested by type II restriction endonucleases (BglII, BamHI, KpnI and XbaI), indicating that the viral genome is a dsDNA molecule. Sequencing revealed that the SFV1 genome contains 37,311 bp, including

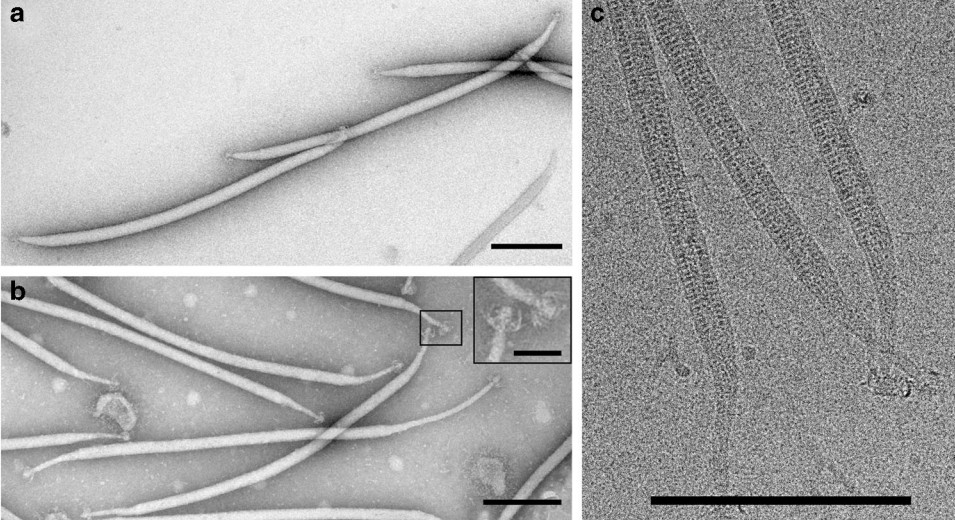

**Fig. 1** Electron micrographs of SFV1 virions. **a** Negatively stained purified virions. **b** Negatively stained virions with an elongated "neck;" mop-like terminal structures are shown in the inset. **c** Cryo-EM of SFV1. Scale bars, 200 nm (50 nm in the inset, **b**)

the 337 bp-long terminal inverted repeats, and has a G + C content of 35.8%. The SFV1 genome contains 66 predicted open reading frames (ORFs) larger than 35 codons that are tightly arranged and occupy 95% of the genome. Fifty-six ORFs use AUG as a start codon.

Homology searches using the BLASTP program[15] revealed that only 12 (~18%) SFV1 gene products are significantly similar (E-value cutoff of 1e-03) to sequences in the non-redundant protein database (Supplementary Table 1). Eight ORFs have homologs in viruses infecting host of the order *Sulfolobales*. One of them, ORF40, has homologs in members of the viral families *Lipothrixviridae*, *Spiraviridae*, *Ampullaviridae* and *Portogloboviridae*. Another, ORF28, has homologs in the *Lipothrixviridae* and *Rudiviridae*, while the remaining six (ORFs 2, 44, 48, 49, 50, 52) are exclusively shared with members of the *Lipothrixviridae* (Supplementary Table 1).

A combination of BLASTP analysis and a more sensitive hidden Markov model (HMM)-based HHpred[16] analysis allowed the assignment of putative functions to less than one-fifth of all SFV1 ORFs (12 ORFs, 18%) (Fig. 2a and Supplementary Table 1). Such resistance to functional annotation based on sequence analyses appears to be a general trend among hyperthermophilic archaeal viruses[17]. Six ORFs, namely ORFs 10, 16, 19, 40, 51, and 65, were predicted to be involved in carbohydrate metabolism. The former five encode putative glycosyltransferases, which may be involved in glycosylation of the viral and/or host proteins,

whereas ORF65 encodes a putative GDP-mannose 4,6-dehydratase (HHpred probability of 100). Two ORFs, ORF2 and ORF52, encode divergent superfamily 2 helicases, both having homologs in lipothrixviruses (Supplementary Table 1). ORFs 26 and 57 are predicted to encode DNA-binding proteins with ribbon-helix-helix (ORF26) and zinc finger (ORF57) domains, respectively. ORF11 encodes a protein with an N-terminal Bergerat-fold ATPase domain of the GHKL (gyrase, heat-shock protein 90, histidine kinase, MutL) superfamily, which is fused to a domain of unknown function and provenance (Supplementary Table 1). The gene product of ORF58 is a Cas4-like nuclease, homologs of which are relatively widespread in archaeal viruses[18], including rudivirus SIRV2, where the protein has been experimentally characterized and suggested to participate in certain stages of genome replication and repair[19] or degradation of the host chromosome[20]. ORF1 and ORF66 are encoded within the terminal inverted repeats and, accordingly, are identical to each other (Fig. 2a). In the rudivirus SIRV2, the genes located within the terminal inverted repeats are the first to be expressed[21] and encode proteins believed to recruit the cellular replisome for viral genome replication via interaction with the host-encoded PCNA, the DNA sliding clamp[22]. Based on the overall virion morphology and gene content analysis, SFV1 appears to be evolutionarily related to lipothrixviruses. Surprisingly, however, neither of the two major coat proteins, a signature of the *Lipothrixviridae*, could be identified in SFV1 by sequence similarity searches.

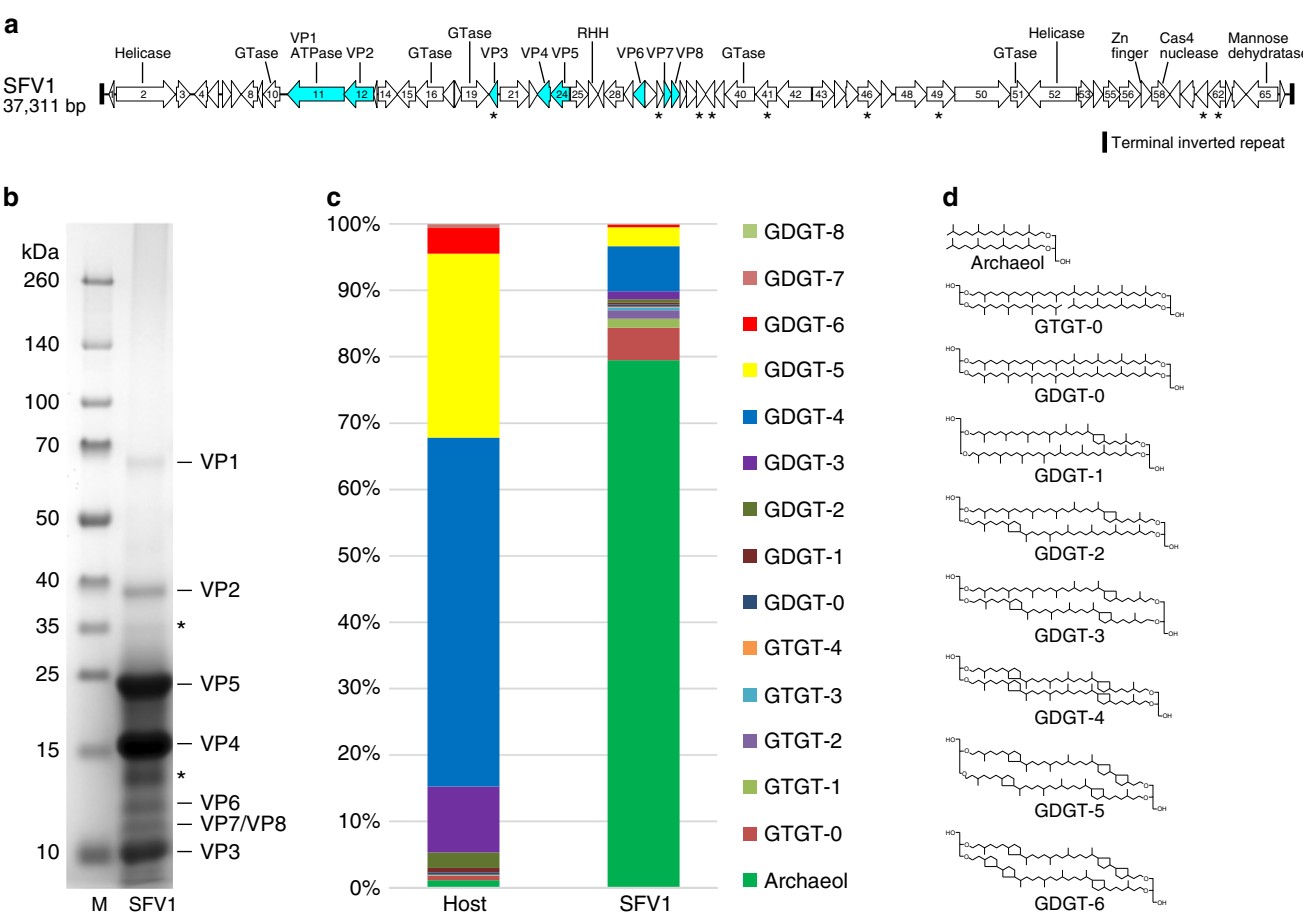

**Fig. 2** Genes, proteins, and lipids of SFV1. **a** Genome map of SFV1. The ORFs are represented with arrows, indicating the direction of transcription. Genes encoding structural proteins are in blue, and ORFs containing predicted transmembrane domains are indicated by asterisks. **b** SDS-PAGE of SFV1 virion proteins stained with Coomassie brilliant blue. The VPs which were identified in proteins bands are indicated. The protein bands marked by asterisks contained a mixture of different VPs in comparable proportion. **c** Distribution of lipid species identified in *S. shibatae* BEU9 cells (Host) and SFV1 virions. **d** Chemical diagrams for the lipid species identified in SFV1 virions and *S. shibatae* BEU9 cells. M molecular mass standards

**Virion composition.** Two prominent bands of equal intensity and seven weaker bands were observed on an electropherogram of highly purified SFV1 virions (Fig. 2b). The viral proteins (VPs) in each band were analyzed by liquid chromatography and tandem mass spectrometry (LC-MS/MS). The results revealed that the two most prominent bands contained proteins VP4 and VP5, with apparent molecular masses of about 16 and 23 kDa, which are encoded by ORF23 and ORF24, respectively (Fig. 2b, Supplementary Table 2). The traces of the two major VPs were detected across the whole length of the gel, most likely due to the presence of their multimers, and/or post-translational modifications. The weaker bands contained minor virion proteins encoded by ORF11 (VP1), ORF12 (VP2), ORF20 (VP3), ORF30 (VP6), ORF33 (VP7), and ORF34 (VP8) (Fig. 2b and Supplementary Table 2). The two bands, which are not labeled in Fig. 2b, contained a mixture of diverse VPs in comparable abundance. Notably, VP1 (ORF11) is a putative Bergerat-fold ATPase, which may be involved in folding of the virion proteins or silencing of the host genes, as has been shown for Hsp90 chaperones and Microrchidia family ATPases, respectively, which also belong to the GHKL superfamily[23,24]. Finally, VP3 contains two putative membrane-spanning α-helical segments, suggesting that it may be embedded within or associated with the virion envelope. We have determined that VP3, VP4, and VP5 are all glycosylated (Supplementary Fig. 2).

Like most archaeal filamentous viruses[7], SFV1 contains an envelope. The presence of lipids in the viral envelope has been confirmed by LC-MS/MS analysis, and the corresponding lipid composition compared to that of the host cell membrane. The distribution of the lipid species in virions is markedly different from that in the host (Fig. 2c). Whereas the cellular membrane is dominated (99% of lipid content) by membrane-spanning $C_{40}$ glycerol dibiphytanyl glycerol tetraether (GDGT) species, particularly GDGT-4 (containing four cyclopentane rings) and GDGT-5, the viral membrane is specifically enriched (80% of the lipid content) in the $C_{20}$ sn-2,3-glycerol diphytanyl ether lipid, also known as archaeol (Fig. 2c). Furthermore, the glycerol trialkyl glycerol tetraether (GTGT), which represents less than 0.7% of the lipid content in the host, was substantially enriched (5%) in the SFV1 envelope. In contrast, the most dominant lipid in the cellular membrane, GDGT-4, represented only 7% of the viral envelope (Fig. 2c). These results indicate that incorporation of lipids into the SFV1 envelope from the host lipid pool is highly selective. Interestingly, envelopes of other crenarchaeal viruses, such as lipothrixvirus AFV1 (ref [9]) and fusellovirus SSV1 (ref [25]), lack archaeol and instead are enriched in GDGT-0 which, in the case of AFV1, assumes a U-shaped horseshoe conformation[9]. The dominance of the flexible $C_{20}$ archaeol suggests that the SFV1 envelope may be more fluid than that of the structurally related virus AFV1.

**Cryo-EM Structure.** Images of the vitrified virions (Fig. 1c) were analyzed by single-particle methods[26], ignoring the specialized structures at the ends and only focusing on the constant diameter tubes which account for most of the virions. We were able to extract ~500,000 overlapping segments of the tubes from ~600 micrographs. Power spectra of the virion segments (Supplementary Fig. 3) suggested that the helical symmetry could be described as $n + 1/7$ units per turn, where $n$ was an integer between 15 and 20, inclusive. By trial-and-error, the correct symmetry (which was 17.14 units/turn) revealed a largely α-helical protein capsid, while incorrect symmetry values gave rise to density that was uninterpretable. The three-dimensional reconstruction showed a total diameter of ~270 Å, and that the nucleocapsid within the outer membrane (Fig. 3a) had a helical pitch of ~47 Å (Fig. 3b).

There is a hollow lumen that is ~95 Å in diameter. In an initial reconstruction, some density appeared near the center of this lumen that was disconnected from the surrounding helical coil and could not be interpreted. Subsequent analysis (Supplementary Fig. 4) revealed that this density arose from regions near the tapered ends of the virus and must be due to the presence of other proteins at these ends. Since it is likely that these internal structures do not have the helical symmetry of the nucleocapsid, the reconstructed density could not be interpreted in this region. The nucleocapsid helix is right-handed, as determined by the hand of the α-helices that are clearly visualized. The asymmetric unit was shown to contain VP4, VP5 and 12 bp of DNA, since the 3.7 Å resolution of the reconstruction (Supplementary Fig. 5) allowed building an unambiguous atomic model for both VP4 and VP5 (Fig. 4a) as well as for the DNA (Fig. 4b, c). The DNA exists in the A-form, as has been shown for SIRV2[8] and AFV1[9]. Since there are 12 bp per asymmetric unit, with 17.14 units/turn, there are ~120 asymmetric units in seven 47 Å pitch helical turns. This means that there are 1440 bp in 127 turns (120 local right-handed turns and seven right-handed superhelical turns), so the DNA has a twist of 11.3 bp/turn. In SIRV2 there are 11.2 bp/turn, while in AFV1 there are 10.8 bp/turn. These values all bracket 11 bp/turn, the "canonical" twist for A-form DNA[27]. The helical radius of the DNA solenoidal supercoiling in SFV1 is ~75 Å, in contrast to SIRV2 where it is ~60 Å and in AFV1 where it is ~30 Å. Given the similar asymmetric units in all three viruses (see below), this is why there are 17.14 asymmetric units per turn in SFV1, 14.7 in SIRV2 and only 9.3 in AFV1.

An interesting difference between SFV1 and both SIRV2 and AFV1 is that the protein subunits contain N-terminal portions that project into the DNA groove. In VP5 (Fig. 5a) approximately six or seven residues can be seen making a deep insertion into the hollow core of the DNA (in A-DNA there is such a hollow core, which does not exist in B-DNA). Although the backbone density can be visualized very clearly, the absence of sidechain densities suggests that the conformation of these sidechains may be dictated by the local DNA sequence. In VP4 no density exists for the first three residues, but two or three residues at the N-terminus (Fig. 5b) can be seen partially inserting into the DNA. In SIRV2, no density was seen for the first six residues, and it is possible that these might be partially inserted into the DNA and in very different positions, so the density disappears after helical averaging. On the other hand, in the AFV1 heterodimer the first six residues of one subunit and the first seven of the other were not seen, but both of these missing regions appear to be too far from the DNA to be compatible with a significant insertion.

Surprisingly, VP4 and VP5 assemble to form a pseudo-symmetric heterodimer (Fig. 6a, b, c), which wraps around the DNA, in a manner similar to the symmetrical capsid homodimer in SIRV2 (Fig. 6d) and the pseudo-symmetrical heterodimer in AFV1 (Fig. 6e). This is surprising since a BLAST search[15] using the VP5 sequence not only failed to find VP4 as a possible homolog, but also failed to find any other protein in the existing protein databases as having significant sequence similarity. In contrast to the compact dimers in SIRV2 and AFV1, VP5 contains a small C-terminal domain containing two short helices (Fig. 6a, b) that projects away from the dimer. This small VP5 domain makes extensive contacts with a VP4 subunit in an adjacent helical turn (Fig. 3c) as well as with the C-terminal domain from two other VP5 subunits in the same helical turn. It is likely that this small domain limits the axial extensibility of the virion, as AFV1, which lacks this extension, was observed to have a helical pitch that varied from ~39 to 47 Å[9]. SIRV2, which also lacks this extension but is more ordered than AFV1, has a more compact helical pitch of 42 Å, which likely stabilizes the virion. In addition, VP5 has an insertion containing two β-strands

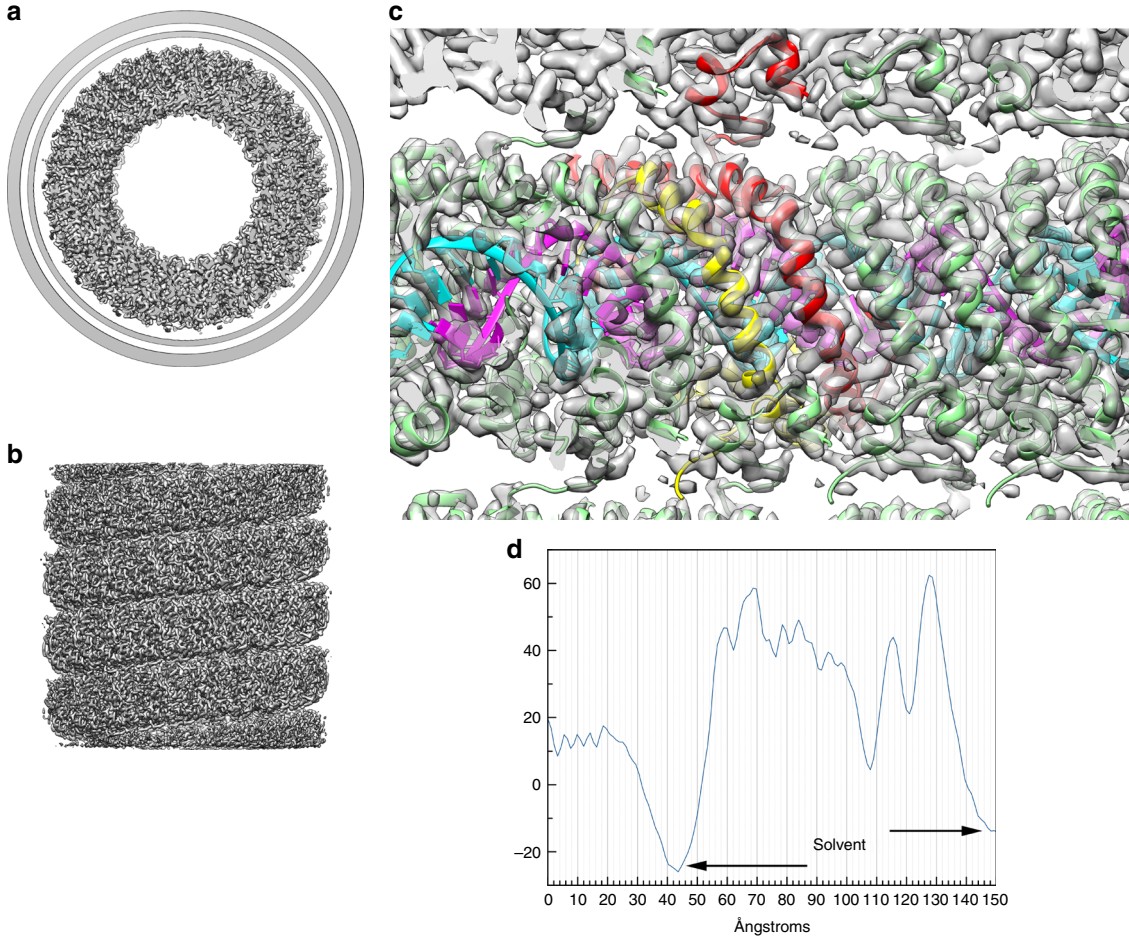

**Fig. 3** Three-dimensional reconstruction of SFV1. A top view (**a**) shows the membrane enveloping the nucleoprotein core. The membrane has been filtered to 7 Å resolution and cylindrically averaged. The nucleoprotein core has been filtered to 3.7 Å resolution. In the side view (**b**) the membrane has been removed. **c** A view from within the lumen of the virion shows the atomic model built into the density. The asymmetric unit contains VP5 (red), VP4 (yellow) and 12 basepairs of DNA. One strand of DNA is shown in magenta, the other in cyan. VP5 contains a small C-terminal domain that makes an extensive contact with a subunit in the helical turn above. **d** The radial density profile of the virion is shown, after cylindrically averaging the structure. Two slightly different estimates of the solvent density come from the inside and the outside of the structure. The higher density on the outside could arise from bound ions or glycosylation, particularly since it has been determined that the membrane protein VP3 is glycosylated (Supplementary Fig. 2)

(Fig. 6b) that is absent in VP4 as well as in the SIRV2 and AFV1 proteins.

While SIRV2 virions are bipolar since they have a twofold axis that is perpendicular to the filament axis and coincides with the local twofold axis in each subunit dimer (Fig. 6d) as well as with the pseudo twofold axis of the DNA (which would be an exact twofold if one ignored the non-palindromic DNA sequence), both SFV1 and AFV1 form polar filaments where the polarity is evident at fairly low resolution. In AFV1 this polarity is due to the pseudo twofold axis of the dimers (Fig. 6e) being tilted by ~26° from the perpendicular to the filament axis. In AFV1, as in SIRV2, the twofold (or pseudo-twofold) axis is perpendicular to the filament, and the overall polarity in SFV1 arises from the asymmetry between the two subunits (Fig. 6c).

**Properties of the membrane**. The reconstruction of the virion (Fig. 3) shows that the membrane has an inner and outer density peak, as might be seen in a normal bilayer. However, as in AFV1, the membrane is anomalously thin, and only ~20–25 Å in thickness. The distance between the inner and outer peaks is 13 Å, in comparison with the 11 Å distance found in AFV1 (ref [9]). As with AFV1, the density of the outer peak is greater than the

inner one, but the density dip between these peaks is larger than in AFV1 and the height of the outer peak is greater than in AFV1 (Fig. 3d). A very crude estimate of the lipid content of the virion, assuming that cryo-EM density is directly proportional to MW (strictly speaking it is not, since the density arises from Coulombic potential), was obtained from simply summing the density in the membrane and comparing it with the total cryo-EM density of the virion. This suggested that the membrane accounts for ~40% of the total mass of the virion. This estimate agrees with a calculation of the membrane mass from simulations described below, which yields an upper limit of 45% of the total mass due to lipids in the cylindrical region of the virion, assuming that all density of the membrane is due to lipids. Since the overall resolution of the SFV1 reconstruction is significantly better than what was achieved for AFV1, the question of whether the membrane is a smooth cylinder (as was stated for AFV1) or is locally perturbed by the helical arrangement of protein subunits can be asked again, and answered with even more confidence. For some icosahedral viruses that contain an internal membrane, it has been shown that the membrane is not spherical, but has an icosahedral shape that is likely maintained by both integral membrane proteins and proteins which span the space between the membrane and the outer capsid[28–30]. What we see for SFV1, as was stated for AFV1,

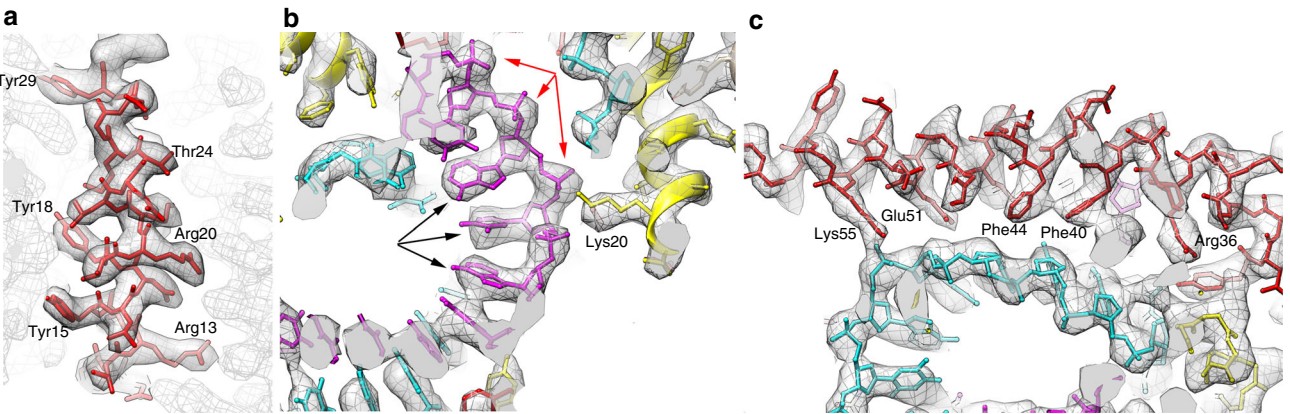

**Fig. 4** The resolution of the map allows for an unambiguous modeling of both the protein and the DNA. **a** A helix from VP5 shows how almost all side chains can be visualized and built into the model. **b** A section of the DNA is shown, with one chain in magenta and the other in cyan. Even though the reconstruction is averaging over the whole genome, most of the bases are clearly resolved (black arrows) and the discrete densities from the phosphate groups are seen (red arrows). A positively-charged side chain from VP4 (Lys20) is adjacent to the negatively-charged phosphate groups. **c** The wrapping of the DNA (cyan) by VP5 (red). Two positively-charged sidechains (Lys55 and Arg36) are in close proximity to the phosphate backbone, while two hydrophobic residues (Phe44 and Phe40) are inserted into the groove in the DNA and are in proximity to the bases

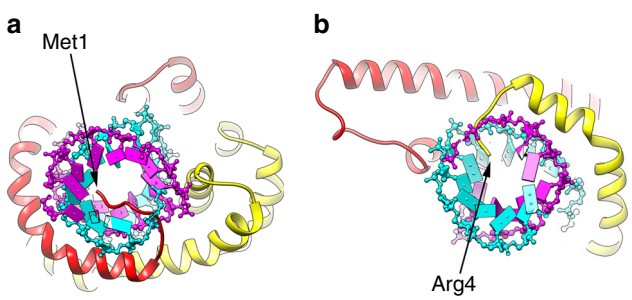

**Fig. 5** N-terminal extensions penetrate the DNA grooves. **a** Six or seven residues at the N-terminus of VP5 (red) project deeply into the DNA. Sidechain density for these residues is absent, suggesting that the conformation of these sidechains may depend on the DNA sequence. **b** Two or three residues from VP4 (yellow) project into the DNA groove. No density for the first three VP4 residues is seen in the reconstruction. One strand of DNA is in magenta, the other in cyan

is that there is no helical perturbation of the membrane by the protein, and that it appears cylindrically symmetric. Since the membrane accounts for ~40% of the mass of the virions, and since there are no features in the averaged power spectrum (Supplementary Fig. 3) arising from any liquid crystalline order of the membrane, all evidence suggests that it is fluid. If we compare the membrane with helical symmetrization with a membrane generated from cylindrically averaging the helical density, we see no systematic differences. This excludes the possibility that the membrane is deformed locally by the protein due to certain amino acid residues facing the membrane or inserted into the membrane.

Simulations of the membrane show that phosphatidylinositol archaeol lipids can reproduce the approximate membrane thickness but cannot account for the large density peaks on the inner and outer edges of the viral membrane (Supplementary Fig. 6). These peaks require a membrane component much more massive than the lipids detected by mass spectrometry. Our best explanation is that some or most of this density may come from VP3, a protein predicted to be nearly all α-helical with two putative membrane-spanning regions. To localize VP3 we have fractionated the virions into a nucleoprotein core and a lipid

envelope (Fig. 7). The analysis of the fractions by SDS-PAGE revealed the presence of VP3 in the envelope fraction (Fig. 7b). The two major capsid proteins, VP4 and VP5, were identified in the fraction containing the nucleoprotein core, which had only negligible amount of VP3 (Fig. 7b).

The density of the outer peak does correspond in width to that expected from α-helices at a radius of 128 Å. These would need to be curved to follow the outer surface of the membrane if oriented in a plane perpendicular to the filament axis, or straight if in a plane parallel to the filament axis, or somewhere in between these two extremes. Given the unknown stoichiometry and structure of VP3, a molecular model of the archaeol:VP3 viral envelope would be grossly underdetermined at this time.

## Discussion

It has been widely recognized that the tertiary structures of proteins can be highly conserved across large evolutionary distances while the primary sequences can diverge considerably[31]. Thus, two proteins that share no detectable sequence similarity can be discovered by structural techniques to have the same fold, and are most likely homologous. Homology can be established more reliably when sequences from a large family of proteins sharing a common fold are known, as in the case, for example, of the actin-hexokinase-HSP70 family[32]. Indeed, comparison of viral structures has provided valuable insights into the origins of viral capsid proteins and revealed deep evolutionary connections in the virosphere that were not apparent from sequence analysis[33,34]. We show here that SFV1 VP4 and VP5, which have no sequence similarity detectable by BLAST[15], are not only extremely similar in fold to each other, but also form a pseudo-symmetrical heterodimer that is very similar to the heterodimer formed by two AFV1 proteins[9] and to the homodimer formed by a single SIRV2 protein[8]. Since all five of these proteins (two in SFV1, two in AFV1, and one in SIRV2) appear to be homologs, the lack of any detectable sequence similarity between, for example, SFV1 VP5 and the other four, suggests that there must be a large number of other sequences sharing the same fold that exist in archaeal viruses that have yet to be discovered or characterized. This leads to the conclusion that the current sampling of viruses that infect acidophilic hyperthermophiles is quite sparse.

The structural similarities observed among SIRV2, AFV1, and SFV1 testify to their common ancestry and that the differences are a

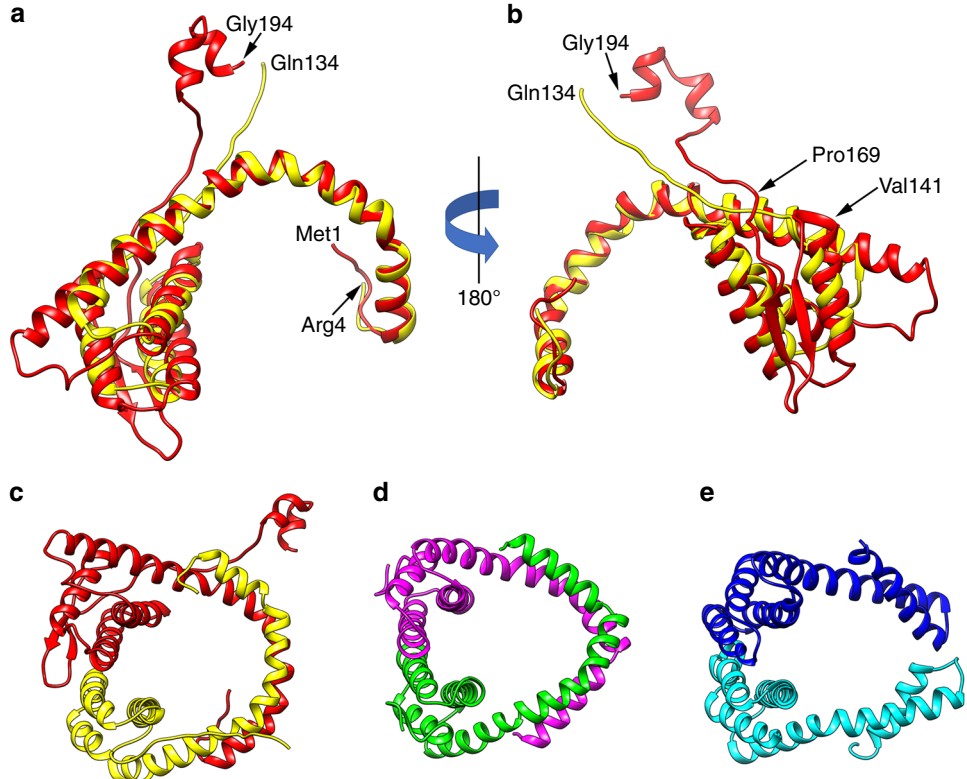

**Fig. 6** Structural conservation and divergence among capsid proteins. Heterodimers (**a**, **b**, **c**, **e**) and homodimers (**d**) form the capsids in SFV1 (**a**, **b**, **c**), SIRV2 (**d**) and AFV1 (**e**). **a**, **b** VP5 (red) and VP4 (yellow) from SFV1 have been aligned to each other. The long α-helix that wraps around the DNA is structurally quite conserved between the two. The main differences between the two coat proteins is that VP5 has an insert (from ~141 to 169) that contains the two β-strands seen in (**b**), and VP5 has a C-terminal extension that contains two short helices. This C-terminal extension crosses the helical groove and makes a large contact with a subunit in the next turn. The SFV1 heterodimer (**c**) is more similar in how it wraps DNA to the SIRV2 homodimer (**d**) than it is to the AFV1 heterodimer (**e**)

result of divergence rather than convergence towards one particular solution among filamentous thermophilic viruses. Thus, they adopt similar structures not because this is the optimal way to build a virus in such aggressive environments but simply because of common ancestry. This is supported by the fact that the virion organization of the evolutionarily unrelated helical virus APBV1[35] is completely different. The latter virus forms a hollow shell into which the circular dsDNA genome is stuffed. The examples of SIRV2, AFV1, and SFV1 are a testament to the fact that, despite deep divergence, evolutionarily related viruses preserve common principles of virion assembly and structure, as has been previously demonstrated for icosahedral viruses with the capsid proteins carrying the double jelly-roll or HK97-like folds[36].

While it is surprising that the two SFV1 proteins form a dimer that is very similar to the DNA-wrapping dimers described for AFV1 and SIRV2, it is perhaps even more surprising that the thin membrane that envelops SFV1 is very similar in profile to the membrane that envelops AFV1 (ref. [9]), even though the component lipids in both are extremely different. We speculate that these membrane properties are required to maintain environmental stability of the virion while permitting viral entry. Enveloped viruses generally have lipids in a liquid crystalline (fluid) phase, while small gel-phase vesicles are not stable and readily fuse to form bilayers on surfaces[37–39]. SFV1 and AFV1 provide examples of how these properties can be achieved using different archaeal lipids that are themselves highly dissimilar to the lipids that predominate in eukaryotic enveloped viruses. This demonstrates yet again that nature has evolved many different ways to accomplish similar functions. Understanding how these

structures are assembled may give us new tools in fields ranging from nanotechnology to materials science.

## Methods

**Enrichment culture and isolation of SFV1 and host strain.** The 10 ml aliquot of the environmental sample was used to inoculate 40 ml of *Sulfolobus* medium of which the components were described by Zillig et al.[40]. The culture was incubated for 10 days at 75 °C under aerobic conditions. Single strains were colony purified from the enrichment cultures by plating on Phytagel (Sigma-Aldrich) plates and incubated for 5 days at 75 °C. To isolate the sensitive host of SFV1, 2 μl aliquot of cell-free enrichment culture supernatant containing VLPs were dropped onto the Phytagel plate that contained a mixture of soft layer and cells of each isolate. After 3 days at 75 °C, halo zones around the drops were observed on cell lawns of several isolates. The halo zones were then inoculated to exponentially growing cultures of the corresponding isolates. After 2 days at 75 °C, the replication of the virus was monitored by TEM observation of the culture supernatant. The single SFV1 virus strain was isolated from a single plaque formed on the lawn of isolated *S. shibatae* strain BEU9. The primers used for amplification of 16S rRNA gene of *S. shibatae* BEU9 were A21F (5′-TTCCGGTTGATCCYGCCGGA-3′) and U1525R (5′-AAG GAGGTGATCCAGCC-3′).

**SFV1 production and purification.** About 25 ml of exponentially growing *S. shibatae* BEU9 cell culture (OD600: ~0.6) was infected by SFV1 at an MOI of 0.1–0.5. After incubation at 78 °C for 24 h, the infected culture was diluted by eight volumes of fresh medium and one volume of fresh exponentially growing BEU9 cells, and further incubated for 24 h at 78 °C. This procedure was repeated once, until the volume of the infected cell culture reached 2.5 l. After final round of dilution, the cell culture was incubated at 78 °C for 36 h. The cells were removed by centrifugation (Sorvall SLA 3000 rotor, 13,689 × *g*, 30 min). From the cell-free supernatant virus was precipitated by addition of PEG6000 (Sigma-Aldrich) and NaCl until the final concentrations of 10% (wt/vol) and 5.8% (wt/vol), respectively. The virus pellet was collected by centrifugation (Sorvall SLA 1500 rotor, 29,753 × *g*, 40 min) and suspended in about 10 ml of sample buffer containing 20 mM Tris acidic buffer (pH 6) and 3% (wt/vol) NaCl. The concentrated SFV1 virions were

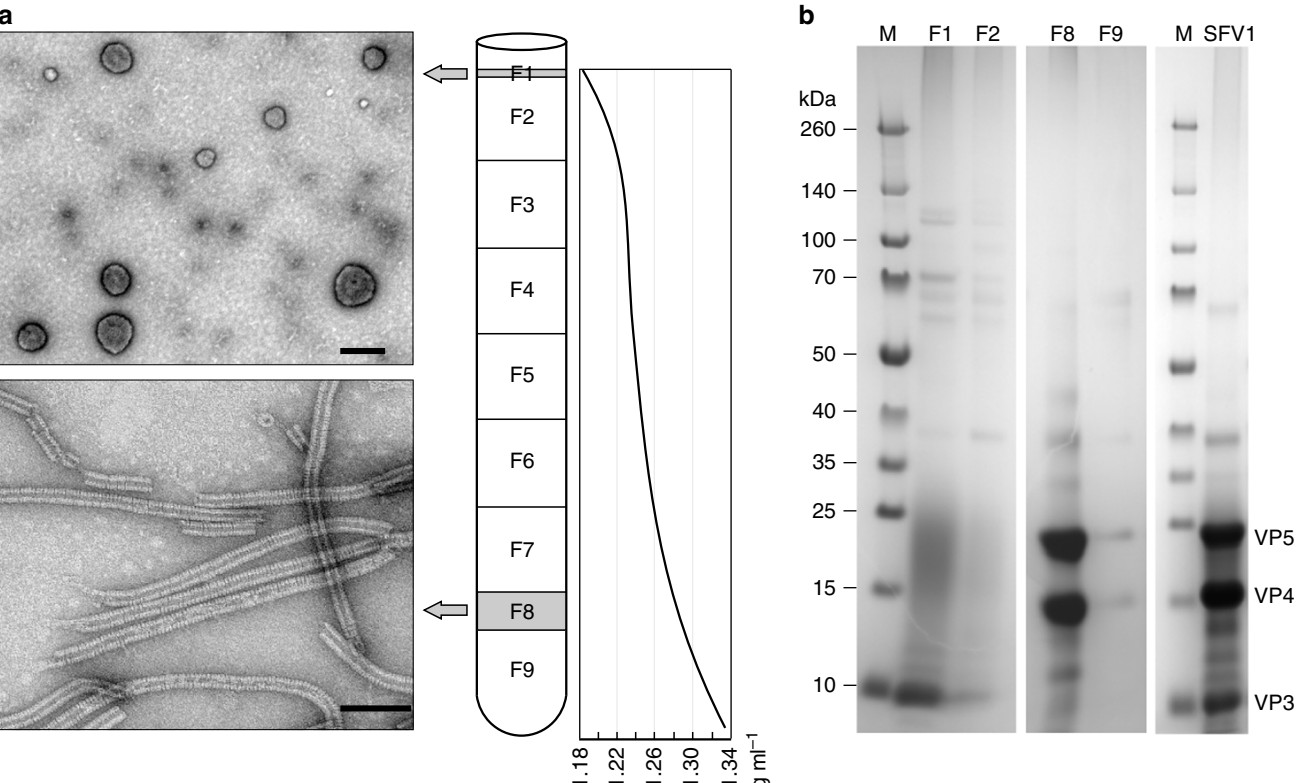

**Fig. 7** Separation of fragments of Triton X-treated SFV1 virions by CsCl gradient cenrifugation. **a** The scheme shows collected fractions and their density, along with electron micrographs of the observed particles from two fractions negatively stained with 2% uranyl acetate. Scale bars: 500 nm in the upper panel, 100 nm in the lower panel. **b** SDS-PAGE of collected fractions after staining with Coomassie brilliant blue; M molecular mass standards, SFV1 SDS-PAGE of intact virions, with indication of proteins VP3, VP4, and VP5

purified in 10–25% sucrose gradient (in the sample buffer) by rate zonal centrifugation (Beckman SW32 Ti rotor, 82,667 × g, 20 min, 15 °C), and the virus light-scattering zone was collected and further purified in 0.45 g ml⁻¹ CsCl (in the sample buffer) by isopycnic gradient centrifugation (Beckman SW60 Ti rotor, 215,000 × g, 17 h, 15 °C).

**Negative-stain electron microscopy**. For negative-stain TEM, 5 μl of the sample was added to a Formvar-coated copper grid (Ted Pella Inc.) and stained with 2% (wt/vol) uranyl acetate. Samples were imaged with an FEI Tecnai Biotwin 120 TEM at 120 kV (Ultrapole of Institut Pasteur, France).

**Extraction and analysis of viral DNA**. Nucleic acid was isolated from the purified virions with phenol/chloroform/isoamyl alcohol (25:24:1 vol/vol). Before extraction, the virus preparation was treated with SDS and Proteinase K with final concentrations of 0.5% (wt/vol) and 0.2 mg ml⁻¹, respectively for 1 h at 55 °C. SFV1 genome library was prepared with the NEBNext Ultra DNA library Prep Kit for Illumina (New England Biolabs), and samples were sequenced by Illumina Miseq with paried-end 150-bp read lengths (Genomics Platform, Institut Pasteur, France). The genome sequence was assembled using the CLC Genomics Workbench software package. An average coverage of 800 was obtained. ORFs were predicted by using RAST v2.0[41]. Homologs of the in silico-translated protein sequences were searched using BLASTP[15] with an upper threshold E value of 1e-3, against the non-redundant protein database at the National Center for Biotechnology Information. Searches for distant homologs were performed by using HHpred[16] against different protein databases (PFAM (Database of Protein Families), PDB (Protein Data Bank), CDD (Conserved Domain Database), and COG (Cluster of Orthologous)). Transmembrane domains were predicted using TMHMM[42].

**Analysis of SFV1 structural proteins**. The highly purified virions were analyzed by 4–12% gradient NuPAGE Bis-Tris precast gel (Thermo Fisher Scientific). Proteins were stained with Coomassie blue using InstantBlue (Expedeon), and the stained protein bands were excised from the gel. Proteins were in-gel tryptic digested, and the generated peptides were analyzed by nano-LC-MS/MS (Proteomics Platform, Institut Pasteur, France) using an Ultimate 3000 system (Dionex) coupled to an LTQ-Orbitrap Velos system (Thermo Fisher Scientific). Peptide masses were searched against annotated SFV1 proteins using Andromeda[43] with

MaxQuant software, version 1.3.0.5[43]. Glycosylation of VPs was detected by using a Pro-Q Emerald 300 glycoprotein gel stain kit (Thermo Fisher Scientific).

**Dissociation of SFV1 virions**. The purified SFV1 virions were treated with Triton X-100 with a final concentration of 0.5% (vol/vol) for 8 h at room temperature. After treatment, the virion fragments were separated by isopycnic gradient centrifugation in CsCl in the presence of 0.3% (vol/vol) Triton X-100. Nine fractions were collected and analyzed by SDS-PAGE, and TEM; prior to TEM observations Triton X-100 was removed from the fractions with Bio-Beads (Bio-Rad). Light scattering bands were observed in fractions F1 and F8. In the fraction F1 were observed pleomorphic spherical particles of different sizes, most likely detached envelopes (Fig. 7a). The nucleoprotein core of the virion was observed in fraction F8 (Fig. 7a).

**Analysis of SFV1 and host cell lipids**. The freeze-dried host-cell preparation and the virion preparation were directly acid hydrolyzed by refluxing with 5% HCl in methanol for 3 h to release GDGT lipids. GDGT lipids were analyzed by high-performance liquid chromatography/atmospheric pressure chemical ionization-mass spectrometry (UHPLC/APCI-MS) using an Agilent 1260 HPLC, equipped with automatic injector, coupled to a 6130 Agilent MSD and HP Chemstation software. The injection volume was 10 μl. Separation of the GDGTs was achieved in normal phase using two silica BEH HILIC columns in series (150 mm × 2.1 mm; 1,7 μm; Waters Acquity) at a temperature of 25 °C. The mobile phases are hexane (A) and hexane:isopropanol (9:1, v-v) (B). Compounds were isocratically eluted for 25 min with 18% B, followed by a linear gradient to 35% B in 25 min and a linear gradient to 100% B in the 30 min thereafter. The flow rate was kept constant (0.2 ml min⁻¹) during the analysis. The mass spectrometer was operated in single ion mode (SIM) to monitor archaeol, GTGTs and GDGTs with 0–8 cyclopentane moieties. Relative abundances of these lipids were determined by integrating peak areas of the SIM signal.

**Cryo-EM and image analysis**. The purified virus preparation (4 μl, 1–2 μg μl⁻¹) was applied to lacey carbon grids that were plasma cleaned (Gatan Solarus) and vitrified in a Vitrobot Mark IV (FEI, Inc.). Grids were imaged in a Titan Krios at 300 keV, and recorded with a Falcon III direct electron detector at 1.09 Å per pixel, with 24 "fractions" per image. Each fraction, containing multiple frames, represented a dose of ~ 2 electrons/Å². All the images were first motion corrected by

MotionCorr v2.1[44], and the first and the last fractions were removed. Then a total of 586 images (each 4k x 4k) were selected that were free from drift or astigmatism, contained visible virus filaments, and had a defocus range from 0.5 to 3.0 μm determined by CTFFIND3[45], The SPIDER software package[46] was used for most subsequent steps. The CTF was corrected by multiplying each image by the theoretical CTF, both reversing phases where they need to be reversed and improving the signal-to-noise ratio. This multiplication of the images by the CTF is actually a Wiener filter in the limit of a very poor SNR. The program e2helixboxer within EMAN2[47] was used for boxing long filaments from the micrographs, and 1972 such boxes of varying length were extracted. Overlapping boxes, 384 px long with a 4 px shift between adjacent boxes (~1.5 times the axial rise per subunit) were extracted from these long filaments, yielding 486,642 segments that were padded to 384 × 384 px. The CTF determination and particle picking came from the integrated images (all 22 fractions after motion correction), while the segments used for the initial alignments and reconstruction came from the first ten fractions.

The determination of the helical symmetry was by trial and error, searching for a symmetry which yielded recognizable secondary structure[48]. Both SPIDER[46] and Relion[49] were used to generate independent reconstructions, which were very similar.

The IHRSR algorithm[26] was used for the SPIDER reconstructions, starting from a solid cylinder as an initial model. Once the correct symmetry was determined (an axial rise of 2.76 Å and a rotation of 21.0° per subunit), the segments were processed by the IHRSR method to produce the reconstructions. Finally, the variability in the structure was further overcome by only symmetrizing the central sixth 32 px of the 384 px long asymmetric reconstruction. Since the images had been multiplied by the CTF twice (once by the microscope and once by us in phase correction), the amplitudes of the final volume were divided by the sum of the squared CTFs. The reconstructed volume (which has a very high SNR from the extensive averaging) is corrected only in Fourier amplitudes by dividing by the sum of the squared CTFs. This is a Wiener filter in the limit of a very high SNR. The map was further sharpened using a negative B-factor of 190.

To further improve the resolution, micrographs with 24 fractions and boxing coordinates were imported into Relion[49]. A similar reconstruction to the one from SPIDER was generated in Relion using the same helical symmetry, starting with the SPIDER volume filtered to 10 Å. The resolution was then further boosted from 4.0 to 3.7 Å in Relion by movie-refinement, dose weighting and particle polishing.

**Model building**. The atomic model of SFV1 was built using the de novo model-building protocol of Rosetta[50]. First, the map corresponding to a single VP4/VP5 asymmetric unit was segmented from the filament density map by Chimera. Then for both VP4 and VP5 a corresponding fragment library that contains pieces of experimentally determined structures was generated from the Robetta server[51]. This allowed for approximately 75% of the backbone of both VP4 and VP5 to be successfully built by Rosetta. Then the full-length VP4/VP5 asymmetric unit was rebuilt with the RosettaCM protocol[50]. For both VP4 and VP5, a total of 3000 full-length models were generated based on the segmented map, and the top ~15 models were selected according to Rosetta's energy function. These selected models were then combined into one model by manual editing in Coot[52] using the criteria of the local fit to the density map and the geometry statistics of the model. The combined model was further refined by Phenix real space protocol to yield single atomic models of VP4/VP5.

Following the procedures from SIRV2[8] and AFV1[9], we also built models for the A-form DNA. The density corresponding to A-form DNA was first segmented from the filament map using Chimera. Then the A-form DNA model was docked into the map and further refined by the Phenix real space protocol to generate the best local fit of model to density. After this, DNA and the VP4/VP5 models were combined to generate a filament model. An additional Phenix real space refinement step was done to generate the final filament model, and for this step the DNA was held fixed as a rigid body so only protein conformation was optimized. MolProbity[53] was used to evaluate the quality of the models, and the statistics are listed in Supplementary Table 3.

**Membrane simulations**. Simulations of the SFV capsid and membrane were performed using Gromacs[54] and the CHARMM36 force field[55], with phosphatidyl-archaeol lipid parameters constructed from trivial modification of our previously published PI-GDGT0 parameters[9]. Lipids were placed in a cylindrical arrangement around the viral capsid, energy-minimized, solvated in TIP3P water and 150 mM NaCl, and then simulated for 450 ns under NVT conditions using the velocity-rescaling thermostat[56] at 4 °C with long-range electrostatics treated using Particle Mesh Ewald[57]. All other simulation parameters were as in our previous work[9]. Simulations were performed at 0, 4, and 37 °C, and the viral membrane was similarly stable in all of them.

**Data availability**. The sequences of the SFV1 genome and the host *Sulfolobus* BEU9 16S rRNA gene have been deposited in Genbank, with accession numbers MH447526 (SFV1) and MH443276 (*Sulfolobus* BEU9 16S). The atomic model of the nucleoprotein complex has been deposited in the Protein Data Bank (6D5F) and the reconstructed volume has been deposited in the Electron Microscopy Data

Bank (EMD-7797). Other data are available from the corresponding authors upon reasonable request.

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

## Acknowledgements

This work was supported by NIH R35GM122510 (to E.H.E.), the European Union's Horizon 2020 research and innovation program under grant agreement 685778, project VIRUS-X (to D.P.) and l'Agence Nationale de la Recherche (France) project ENVIRA (to M.K.). The cryo-EM imaging was conducted at the Molecular Electron Microscopy Core facility at the University of Virginia, which is supported by the School of Medicine and built with NIH grant G20-RR31199. The Titan Krios and Falcon II direct electron detector were obtained with NIH S10-RR025067 and S10-OD018149, respectively. Ellen Hopmans and Caglar Yildiz (NIOZ) are thanked for analytical support.

## Author contributions

D.P. and E.H.E. designed the study. Y.L. isolated and purified SFV1, sequenced the genome and identified the structural proteins. M.K. annotated the genome and performed the bioinformatic analysis. F.W. acquired the cryo-EM images, and E.H.E, F.W., and T.O. generated the reconstruction. T.O. and F.W. generated the atomic model. S.S. analyzed the lipids from both the host and SFV1. P.K. performed the simulations on the viral membrane. D.P., Y.L., M.K., P.K., and E.H.E. wrote the manuscript and all authors commented on the manuscript.

## Additional information

**Competing interests:** The authors declare no competing interests.

