## [Peer Review File · Nature Communications]

Reviewers' comments:

Reviewer #1 (Remarks to the Author):

Viruses that infect archaea living in extreme environments are of interest because of they may help to understand the evolutionary relationship to bacteriophage and eukaryotes viruses and also because of the way these viruses protect their packaged DNA from the damage by extreme environments such as nearly boiling acids.

In this manuscript, Liu et al. isolate Sulfolobus filamentous virus 1 (SFV1) from *S. shibatae* BEU9. The virion morphology and the viral genome were analyzed by negative stain and homology/BLASTP search, respectively. The VPs including the major capsid proteins and the minor capsid proteins that compose the virion were studied by electropherogram and tandem mass spectrometry. The viral membrane is specifically enriched in archaeol, which is different from the host lipid. The virus structure was reconstructed by cryo-EM single particle method to a resolution of 3.7 Å. The atomic models of major capsid protein VP4, VP5 and A-form DNA were built ab-initially according to the density map. Overall, the structure of capsid is similar to SIRV2 and AFV1, two other viruses of archaea. However, some new features were found to from extra contacts between VP4 and VP5 and between VP5 and DNA. The cryo-EM structure is beautiful and the quality of the structure is good enough to build the model.

Major comments,

First, two interesting questions have been raised in the abstract for studying the archaea viruses. One is to understand their evolutionary relationship to viruses that infect bacteria and to those that infect eukaryotes. Another is from the point of view of biotechnology and nanotechnology, as knowing how to package DNA so that it can withstand nearly boiling acid has many potential practical applications. However, the rest part of the manuscript seems not very much related to this two interesting questions. The new structure features which are different from the structures of other two archaea virus solved previously are not seems to help to answer these two questions.

Second, a significant part of the manuscript is about the viral membrane. It is found the component of the membrane of the virus is quite different from the host lipid. It indicates that the incorporation of the host lipid into the SFV1 envelope is highly selective. However, the lipid component of SFV1 is very different from that of AFV1 and SSV1. Thus, the conclusion is the SFV1 envelope may be more fluid than that of other two viruses. In the structure of SFV1, there are no helical perturbation of the membrane by the protein and no features in the averaged power spectrum. It also indicates that the viral membrane is fluid. Is there any specific function of fluid membrane in the virus life cycle for this virus? Since other two viruses do not need the fluid viral membrane, therefore, the function of fluid membrane is probably not related with the common interesting questions in the abstract. The authors used the ratio of the density of membrane to that of protein in cryo-EM map to calculate the ratio of the molecular weight of membrane to that of protein. However, the ratio of the density of membrane to that of protein is quite different at different frequency. For instance, the ratio becomes very small at high frequency but becomes large at low frequency. Therefore, the ratio is very much related to the b factor that was applied to the map. The conclusion that 40% mass of the virus belongs to the membrane has to be considered more carefully. The same caution has to be paid for the simulation of the intensity of the viral membrane.

Reviewer #2 (Remarks to the Author):

Liu et al. describe isolation and characterization of a new filamentous virus SFV1 infecting *Sulfolobus shibatae*. The basic components of the virions i.e. structural proteins, lipid envelope and

the genome sequence are described and the host range was studied using fairly limited number (7) of different strains of *Sulfolobus islandicus*, *S. acidocaldarius* and *S. solfataricus*. In addition, the virion structure was described using cryo-EM. The conclusions are based on sound conduct of experiments by clear professionals of each field, i.e. crenarchaeal and structural biology as well as experts in lipid chemistry yielding a structural description of the virion at 3.7 Å resolution. All of this is done to the point where many of the methods have been described just by referring to previous publications. The study results in observations concluding that although many details can be quite different, the overall picture of SFV1 virion is relatively similar to previously described crenarchaeal SIRV2 and in particular another membrane containing filamentous virus AFV1. In conclusion, the obtained results are not that novel after all. The cryo-EM resolution obtained in this study is better than those of AFV1 (4.1-4.5Å) and SIRV2 (~4Å), but it seems that improvements of resolution alone do not help in finding novelty. Novelty may have been found in the structure, formation and some kind of a significance of the long necks in the ends of the virion, but this was not studied.

If novelty cannot be found, the study could go a bit deeper in certain aspects that have now been left at the level of speculation, although as such providing more data does not necessarily lead to more novelty.

Since there are no line nor page numbers in the manuscript, I hope I am clear about the places of my comments.

I have only few detailed comments:

Abstract

"These viruses are of interest for a number of reasons. One is understanding their evolutionary relationship to viruses that infect bacteria (bacteriophage) and to those that infect eukaryotes."

The authors very shortly mention something about the structure of TMV, but they completely forget filamentous bacteriophages such as M13 or alike. Why?

Results/Viral genome

The authors mention that the genome contains putative ORFs encoding glycosyltransferases and that they may be involved in glycosylation of the viral and/or host proteins. Glycosylation of virion proteins could be shown by staining of a gel containing the structural proteins.

The role of VP3

It is predicted that VP3 contains two membrane spanning alpha-helical segments and some of the density in the membrane could be a result of these membrane domains. If the membrane is removed, what happens to VP3? I wonder whether you can remove the membrane using a method which does not disrupt the nucleoprotein part of the filament thus allowing the structural examination of the nucleoprotein core.

Materials and Methods

I quote the the instructions to the authors:

"Please provide a Data Availability statement in the Methods section under "Data Availability"; detailed guidance can be found in our data availability and data citations policy. Certain data types must be deposited in an appropriate public structured data depository (details are available here) and the accession number(s) provided in the manuscript. Full access is required at publication. Should full access to data be required for peer review, authors must provide it."

This statement is totally missing together with an accession number of the genomic sequence submitted to a sequence database.

Reviewer #3 (Remarks to the Author):

This work reported in this manuscript is well done and the major conclusions of the work are well supported by the presented results/data. There are three significant findings: 1) the report of a new archaeal virus, and its genome sequence 2) the understanding on how two structural proteins with little primary sequence homology form a heterodimer that interacts with dsDNA to form A DNA and how a very similar fold is utilized by two other archaeal viruses (but with very different proteins) to package A form DNA in their virions, 3) and the characterization of a 'fluid' non-helical external lipid membrane. These are exciting findings that will be of interest to a broad range of life scientists.

The manuscript could be further improved by the following: 1) the addition of more structural details on the connection (or lack of connections) between the lipid envelope and the interior nucleocapsid structure and 2) include in the Discussion a section that places the SFV1 structure in a more biological context (why are filamentous thermophilic viruses adopting similar structures?).

Overall, an excellent piece of work worthy of publication in Nature Communications.

Reviewer #1 (Remarks to the Author):

Viruses that infect archaea living in extreme environments are of interest because of they may help to understand the evolutionary relationship to bacteriophage and eukaryotes viruses and also because of the way these viruses protect their packaged DNA from the damage by extreme environments such as nearly boiling acids.

In this manuscript, Liu et al. isolate Sulfolobus filamentous virus 1 (SFV1) from *S. shibatae* BEU9. The virion morphology and the viral genome were analyzed by negative stain and homology/BLASTP search, respectively. The VPs including the major capsid proteins and the minor capsid proteins that compose the virion were studied by electropherogram and tandem mass spectrometry. The viral membrane is specifically enriched in archaeol, which is different from the host lipid. The virus structure was reconstructed by cryo-EM single particle method to a resolution of 3.7 Å. The atomic models of major capsid protein VP4, VP5 and A-form DNA were built ab-initially according to the density map. Overall, the structure of capsid is similar to SIRV2 and AFV1, two other viruses of archaea. However, some new features were found to from extra contacts between VP4 and VP5 and between VP5 and DNA. The cryo-EM structure is beautiful and the quality of the structure is good enough to build the model.

Major comments,

First, two interesting questions have been raised in the abstract for studying the archaea viruses. One is to understand their evolutionary relationship to viruses that infect bacteria and to those that infect eukaryotes.

We have clarified this point in the paper, as simply understanding the evolutionary relationship between SFV1 and both SIRV2 and AFV1 (since we have established that all three are homologs) will require studies of a vast family of other filamentous archaeal viruses that we predict must exist. We expect that this will provide a better basis for then understanding the possible relations with bacterial and eukaryotic viruses.

Another is from the point of view of biotechnology and nanotechnology, as knowing how to package DNA so that it can withstand nearly boiling acid has many potential practical applications. However, the rest part of the manuscript seems not very much related to this two interesting questions. The new structure features which are different from the structures of other two archaea virus solved previously are not seems to help to answer these two questions.

We agree that we have not answered either of these questions. But we think the reviewer would agree that the paper makes a significant contribution by expanding our knowledge in this field. The two questions will continue to be of interest and will motivate a substantial amount of further work in this area.

Second, a significant part of the manuscript is about the viral membrane. It is found the component of the membrane of the virus is quite different from the host lipid. It indicates that the incorporation of the host lipid into the SFV1 envelope is highly selective. However, the lipid component of SFV1 is very different from that of AFV1 and SSV1. Thus, the conclusion is the SFV1 envelope may be more fluid than that of other two viruses.

We agree with this, but AFV1 also appeared fluid (see below).

In the structure of SFV1, there are no helical perturbation of the membrane by the protein and no features in the averaged power spectrum. It also indicates that the viral membrane is fluid.

This was also true for AFV1, but we can establish this at higher resolution for SFV1.

Is there any specific function of fluid membrane in the virus life cycle for this virus? Since other two viruses do not need the fluid viral membrane, therefore, the function of fluid membrane is probably not related with the common interesting questions in the abstract.

We have added several sentences suggesting a role for the fluidity, and added three new references in support.

The authors used the ratio of the density of membrane to that of protein in cryo-EM map to calculate the ratio of the molecular weight of membrane to that of protein. However, the ratio of the density of membrane to that of protein is quite different at different frequency. For instance, the ratio becomes very small at high frequency but becomes large at low frequency. Therefore, the ratio is very much related to the b factor that was applied to the map. The conclusion that 40% mass of the virus belongs to the membrane has to be considered more carefully. The same caution has to be paid for the simulation of the intensity of the viral membrane.

We agree that this was a very crude calculation. We have added a sentence that the simulation gives a value of 45% of the total mass, but also added a caveat for this estimate. We have further emphasized that both estimates are very crude, but that both suggest that the membrane must account for a substantial fraction of the total mass.

Reviewer #2 (Remarks to the Author):

Liu et al. describe isolation and characterization of a new filamentous virus SFV1 infecting *Sulfolobus shibatae*. The basic components of the virions i.e. structural proteins, lipid envelope and the genome sequence are described and the host range was studied using fairly limited number (7) of different strains of *Sulfolobus islandicus*, *S. acidocaldarius* and *S. solfataricus*. In addition, the virion structure was described using cryo-EM. The conclusions are based on sound conduct of experiments by clear professionals of each field, i.e. crenarchaeal and structural biology as well as experts in lipid chemistry yielding a structural description of the virion at 3.7 Å resolution. All of this is done to the point where many of the methods have been described just by referring to previous publications. The study results in observations concluding that although many details can be quite different, the overall picture of SFV1 virion is relatively similar to previously described crenarchaeal SIRV2 and in particular another membrane containing filamentous virus AFV1. In conclusion, the obtained results are not that novel after all. The cryo-EM resolution obtained in this study is better than those of AFV1 (4.1-4.5Å) and SIRV2 (~4Å), but it seems that improvements of resolution alone do not help in finding novelty. Novelty may have been found in the structure, formation and some kind of a significance of the long necks in the ends of the virion, but this was not studied.

We disagree with the reviewer about novelty. When one of us told Bill Pearson, the co-creator of the FastA sequence alignment algorithm, that we had protein sequences (VP4 and VP5) from SFV1 that showed no significant sequence similarity using BLAST with any other proteins (or with each other) in the entire universe of bacterial, archaeal and eukaryotic proteins, he said that this was impossible, based upon the premise that no "singleton" sequences exist in biology. He then ran the search and said that it was true. What we have been able to show in the paper is that SIRV2, AFV1 and SFV1 are all structural homologs, must have a common ancestor, and therefore hundreds or thousands of other such viruses must exist. Thus, parts of the biosphere on earth have been very, very sparsely sampled. This is indeed a novel conclusion.

If novelty cannot be found, the study could go a bit deeper in certain aspects that have now been left at the level of speculation, although as such providing more data does not necessarily lead to more novelty.

Since there are no line nor page numbers in the manuscript, I hope I am clear about the places of my comments.

I have only few detailed comments:

Abstract

“These viruses are of interest for a number of reasons. One is understanding their evolutionary relationship to viruses that infect bacteria (bacteriophage) and to those that infect eukaryotes.”

The authors very shortly mention something about the structure of TMV, but they completely forget filamentous bacteriophages such as M13 or alike. Why?

We did not forget M13 and similar filamentous phage. We did not mention them because they have circular genomes, and cannot have the type of stoichiometric interaction with capsid proteins found for linear single stranded RNA in TMV or for linear dsDNA in SIRV2, AFV1 and SFV1.

Results/Viral genome

The authors mention that the genome contains putative ORFs encoding glycosyltransferases and that they may be involved in glycosylation of the viral and/or host proteins. Glycosylation of virion proteins could be shown by staining of a gel containing the structural proteins.

This is a good point, and we have now done this (Supp. Fig. 6), and can show that VP3, VP4 and VP5 are glycosylated.

The role of VP3

It is predicted that VP3 contains two membrane spanning alpha-helical segments and some of the density in the membrane could be a result of these membrane domains. If the membrane is removed, what happens to VP3? I wonder whether you can remove the membrane using a method which does not disrupt the nucleoprotein part of the filament thus allowing the structural examination of the nucleoprotein core.

We cannot remove the membrane without perturbing the nucleoprotein core, as shown for AFV1, where removal of the membrane led to a great increase in the flexibility of the nucleoprotein core. We stated in the AFV1 paper: “Since the membrane, which has fluid-like properties, is unlikely to be directly responsible for the increased rigidity of the enveloped virions, it suggests that the presence of the membrane constrains the protein and thus indirectly imparts rigidity to the structure.” But the reviewer raises a good point about the location of VP3. We have now done the experiments where the membrane fraction can be separated from the nucleoprotein core (Fig. 7) and can show that VP3 is associated with the membrane fraction.

Materials and Methods

I quote the the instructions to the authors:

“Please provide a Data Availability statement in the Methods section under “Data Availability”; detailed guidance can be found in our data availability and data citations policy. Certain data types must be deposited in an appropriate public structured data depository (details are available here) and the accession number(s) provided in the manuscript. Full access is required at publication. Should full access to data be required for peer review, authors must provide it.”

This statement is totally missing together with an accession number of the genomic sequence submitted to a sequence database.

We have added such a statement and have included the accession number for the genome.

Reviewer #3 (Remarks to the Author):

This work reported in this manuscript is well done and the major conclusions of the work are well supported by the presented results/data. There are three significant findings; 1) the report of a new archaeal virus, and its genome sequence 2) the understanding on how two structural proteins with little primary sequence homology form a heterodimer that interacts with dsDNA to form A DNA and how a very similar fold is utilized by two other archaeal viruses (but with very different proteins) to package A form DNA in their virions, 3) and the characterization of a 'fluid' non-helical external lipid membrane. These are exciting findings that will be of interest to a broad range of life scientists.

The manuscript could be further improved by the following; 1) the addition of more structural details on the connection (or lack of connections) between the lipid envelope and the interior nucleocapsid structure

Given that we have been unable to establish any connection between the lipid envelope and the nucleocapsid, we are not able to add any details. We see no difference in this regard between using a cylindrically averaged density for the membrane (which would blur out such connections) and the helical reconstruction of the membrane (which would preserve any contacts between the capsid dimers and the membrane, if they existed). We are currently working on an archaeal icosahedral virus where we clearly see such connections between outer capsid proteins and an internal membrane, consistent with other icosahedral virus structures that we cite.

and 2) include in the Discussion a section that places the SFV1 structure in a more biological context (why are filamentous thermophilic viruses adopting similar structures?).

We have now emphasized that structural similarities observed among SIRV2, AFV1 and SFV1 testify to their common ancestry and that the differences are a result of divergence rather than convergence towards one particular solution among filamentous thermophilic viruses. Thus, they adopt similar structures because of common ancestry! This is supported by the fact that virion organization of the evolutionarily unrelated helical virus APBV1¹ is completely different. The latter virus forms a hollow shell into which the circular dsDNA genome is stuffed. The examples of SIRV2, AFV1 and SFV1 are a testament to the fact that, despite deep divergence, evolutionarily related viruses preserve common principles of virion assembly and structure, as has been previously demonstrated for icosahedral viruses with the capsid proteins carrying the double jelly-roll or HK97-like folds.

Overall, an excellent piece of work worthy of publication in Nature Communications.

References

- 1 Ptchelkine, D., Gillum, A., Mochizuki, T., Lucas-Staat, S., Liu, Y., Krupovic, M., Phillips, S. E. V., Prangishvili, D. & Huiskonen, J. T. Unique architecture of thermophilic archaeal virus APBV1 and its genome packaging. *Nature communications* **8**, 1436, (2017).